# Concentrated Pre-Vulcanized Natural Rubber Latex Without Additives for Fabricating High Mechanical Performance Rubber Specimens via Direct Ink Write 3D Printing

**DOI:** 10.3390/polym17030351

**Published:** 2025-01-28

**Authors:** Lin Liu, Jizhen Zhang, Zirong Luo, Na Kong, Xu Zhao, Xu Ji, Jihua Li, Shenbo Huang, Pengfei Zhao, Shuang Li, Yanqiu Shao, Jinlong Tao

**Affiliations:** 1Heilongjiang Key Laboratory of Photoelectric Functional Materials, College of Chemistry and Chemical Engineering, Mudanjiang Normal University, Mudanjiang 157011, China; liulin990705@163.com (L.L.); 18345260314@163.com (X.J.); 2Hainan Provincial Key Laboratory of Natural Rubber Processing, Agricultural Products Processing Research Institute, Chinese Academy of Tropical Agricultural Sciences, Zhanjiang 524001, China; quickly3000@126.com (J.Z.); luozirong3@163.com (Z.L.); kongna109101@126.com (N.K.); zx19960211@163.com (X.Z.); a641167221@163.com (S.H.); pengfeizhao85ac@163.com (P.Z.); shuangl163163@163.com (S.L.); 3Rubber Research Institute, Chinese Academy of Tropical Agricultural Sciences, Haikou 571101, China; foodpaper@126.com

**Keywords:** concentrated pre-vulcanized natural rubber latex, superabsorbent polymer beads, additive-free, high mechanical performance, direct ink writing

## Abstract

Direct ink writing (DIW) is an economical, straightforward, and relatively energy-efficient 3D printing technique that has been used in various domains. However, the utilization of rubber latex for DIW remains limited due to its high fluidity and inadequate support, which makes it challenging to meet the required ink rheological characteristics for DIW. In this study, a concentrated pre-vulcanized natural rubber latex (CPNRL) ink with a high solid content of 73% without additives is developed for DIW 3D printing. The CPNRL ink is concentrated using superabsorbent polymer (SAP) beads, which demonstrates good colloidal stability, favorable rheological properties, and superior printability. The impact of printing angles on the mechanical properties of the rubber specimens based on the CPNRL-73 ink is explored in detail, wherein the tensile strength of the specimen printed at a 90° angle reaches an impressive 26 MPa and a strain of approximately 800%, which surpasses the majority of 3D-printed rubber latex specimens. Additionally, the CPNRL ink can be used to print a wide range of intricate shapes, demonstrating its advantages in excellent formability. The preparation of 3D printable ink using the absorption method will expand the application of elastomers in fields such as customized flexible sensing and personalized rubber products.

## 1. Introduction

In contrast to conventional manufacturing processes, 3D printing enables swift and precise model creation [1,2,3,4] as well as facilitates a reduction in material waste, the optimization of shape, and an improvement in production efficiency [5,6,7]. The 3D printing of elastomers has garnered significant attention in recent years, particularly within the domains of bionic organs, soft robotics, and intelligent healthcare [8,9]. Natural rubber, as a bio-based elastomer, exhibits exceptional elasticity, high tensile strength, and superior tensile recovery [10,11]. These properties render it indispensable across various domains [12]. The swift advancement of 3D printing technology paves a novel pathway for the development and advancement of natural rubber [13,14,15,16,17,18,19,20].

Currently, rubber is mainly printed by fused deposition modeling (FDM) [21,22,23,24], by vat photo-polymerization (VPP) [1,25,26,27] using photo-curable precursors, and by direct ink writing (DIW) [28,29,30]. They have been used to successfully fabricate various complicated elastic rubber structures. However, they also demonstrate some limitations. For example, the 3D printing of FDM needs a much higher temperature, preventing its wide application in rubber and restraining the use of additives that are not resistant to high temperatures. For the VPP technique, toxic photo-initiators are required. Regarding the DIW technique, it is only necessary to regulate the rheological properties of the ink, for instance, by introducing rheological modifiers to satisfy specific requirements. In comparison with the two aforementioned methods, DIW exhibits lower costs and simpler operational procedures [28]. In summary, compared to FDM and VPP, the DIW 3D printing of rubber elastomers is regarded as one of the most promising technologies. It eliminates the need for high temperatures and photosensitizers while enabling the incorporation of a wider range of additives in 3D printing inks. However, the addition of substances such as rheological regulators may result in increased costs and volume shrinkage of printed samples. Moreover, these additives may compromise the mechanical properties of the printed objects. Consequently, there is an urgent need for an additive-free and scalable method to manufacture rubber samples with superior mechanical properties through direct ink writing 3D printing.

In this paper, a new type of CPNRL ink with a high solid content and without the addition of any thickening agents was fabricated through the absorption concentration method based on SAP beads, which is specifically suitable for DIW 3D printing (Figure 1). The colloidal and rheological properties of the CPNRL ink were systematically analyzed. The optimal printing conditions were identified by evaluating a range of extrusion pressures and linear velocities, and the printing accuracy was also explored. Furthermore, the mechanical properties of the printed specimens can be adjusted by raster orientation, and the tensile strength reached an impressive 26.4 MPa when printed in the 90°direction, accompanied by an elongation at break close to 800%. Its mechanical properties surpass the film prepared using the casting method and the majority of 3D-printed specimens based on rubber latex. Additionally, the CPNRL ink can be used to print a wide variety of complex structures and shapes with high precision. The exceptional mechanical properties of natural rubber are fully retained, providing new insights into the processing of complex natural rubber structures with high mechanical performance.

## 2. Materials and Methods

### 2.1. Materials

Concentrated natural rubber latex (60% in dry rubber content) was obtained from GuangKen Group Company in Guangdong (Yangjiang, China). Zinc oxide (ZnO), sulfur, zinc diethyl dithiocarbamat (ZDC), potassium oleate, sodium fluorosilicate, formaldehyde, antiaging agent-264, diffusible agent-N, casein, and 2-mercaptobenzene were purchased from Shanghai Aladdin Biochemical Technology Co., Ltd. (Shanghai, China). The commercial SAP beads were procured from Dinghao Corporation (Yiwu, China). The extrusion 3D bio-printer EFL-BP-6601 was purchased from Suzhou Yong qinquan Intelligent Equipment Co., Ltd. (Suzhou, China), and all the printed specimens in this article were completed by this equipment.

### 2.2. Preparation of PNRL and CPNRL

Pre-vulcanized natural rubber latex (PNRL) is usually fabricated using concentrated natural rubber latex and a curing system that includes sulfur, an accelerator, and an activator. According to the formulation in Appendix A, the mixture was poured into a ball milling tank and milled at 40 rpm·min^−1^ for 72 h to obtain a 50% mass fraction of the vulcanized hybrid dispersion. It is important to note that the dispersion should be filtered through a sieve to remove any bulk material and foam before use. The stabilizer and vulcanizing dispersant were slowly added to the concentrated natural rubber latex while stirring, and the mixing ratios are shown in Appendix A. After the addition, the mixed system was heated and stirred at 52~60 °C for 3 h. At the end of the reaction, the rubber temperature was cooled to room temperature, and PNRL with 58% solids was obtained.

Concentrated pre-vulcanized natural rubber latex (CPNRL) was prepared as follows: 3 wt% SAP beads were slowly added into PNRL with low-speed stirring at 180 rpm·min^−1^. By regulating the duration of the concentration process, CPNRL with a solid content ranging from 62% to 73% can be achieved. CPNRL-62, CPNRL-65, CPNRL-69, CPNRL-71, and CPNRL-73 represent the solid contents of CPNRL of 62%, 65%, 69%, 71%, and 73%, respectively.

### 2.3. Scale Accuracy of 3D-Printed Rubber Specimens

All 3D-printed models were processed using Slic3r software (V1.2.3 (230313)). The design of the rectangular tracing model comprised a length of 20 cm and a width of 1 cm (Appendix A). The height of the gap between the tip of the nozzle and the substrate was set to 90% of the diameter of the nozzle, which was 21 G (inner diameter 0.51 mm). The linear speed was fixed at 1000 mm/min, and the pressure was controlled from 35 to 55 kPa. Additionally, the extrusion pressure was fixed at 45 kPa, and the linear speed was from 100 to 2000 mm/min.

For accuracy testing, the cuboid structures were 3D-printed using a 1 × 2 × 3 cuboid model integrated into the Slic3r software (V1.2.3 (230313)). These structures were printed with a 21 G nozzle head at 100% infill density. Five cube samples were printed for testing accuracy, and the average values were used for analysis.

### 2.4. Preparation of the 3D-Printed Tensile Model and Film Using the Casting Method Based on CPNRL-73

The 3D-printed specimen based on CPNRL-73 for studying tensile properties was designed to be a standardized dog bone tensile model with 100% packing density. Afterward, the printed specimens were vulcanized at 90 °C in an oven for 4 h. For comparison, the CPNRL-73 film was prepared using the casting method. In accordance with the GB/T 18011-2008 (ISO 498:1992, IDT) standard [31,32], CPNRL-73, which initially had a solid content of 73%, was diluted to achieve a solid content of 62% to prepare the casting film. It was filtered using a 200-mesh screen and then casted into the glass plate to keep static conditions for 48 h at normal temperature. Finally, it was heated in an oven at 35 °C for 48 h to complete the vulcanization.

### 2.5. Printing of Complex Rubber Structure Based on CPNRL-73 Ink

The models of letters, simplified tire treads, associated rotating components, and a hollow air rubber bag were created using Slic3r software. Subsequently, these models were fabricated via the CPNRL ink. Finally, the printed complex structures underwent a four-hour heat treatment at 90 °C to complete the vulcanization.

### 2.6. Characterization

The particle size and Zeta potential of PNRL and CPNRL were determined by using Malvern Zetasizer Nano ZS90 (Malvern Instruments Ltd., Worcestershire, UK). Scanning electron microscopy (HITACHI S4800, Tokyo, Japan) was used to study the morphology of the rubber particles present in PNRL and CPNRL and the surface and cross-section morphology of the 3D-printed specimens. TPNRL and CPNRL were diluted 50 times and stained with a 2 wt% osmium tetroxide aqueous solution before testing. The viscoelastic behavior of CPNRL with various solid contents was measured using a 20 mm plate device on a Rheometer (Thermo Scientific™ HAAKE™ MARS, Waltham, MA, USA) at room temperature. The steady-state mode was employed, with shear rates ranging from 0.01 to 1000 s^−1^. The storage modulus and loss modulus were determined in the vibration mode at an angular frequency of 10 rad·s^−1^. The range of shear stress applied was from 0 to 1000 Pa. The tensile tests of the casting rubber film and 3D-printed specimens were conducted utilizing an electronic universal testing machine (ETM103C). Each sample underwent five sets of parallel experiments, and the average values were used for analysis. According to GB/T 528-2009 [33], a dog-bone-shaped specimen was selected and tested at a speed of 500 mm/min to obtain the stress–strain curve.

The equilibrium swelling method was employed to determine the cross-linking density of the vulcanized sample. After precisely weighing 0.2 g of the vulcanized sample, we added 100 mL of toluene solution and stored it in darkness for 7 days. After removing the swollen sample’s surface-attached toluene solution, its weight was measured. The Flory–Rehner formula (1a)–(1c) was utilized to calculate the cross-link density of several groups of samples.(1a)$$\Phi_{r}=\frac{m_{1}/\rho_{p}+m_{2}/\rho_{s}}{m_{1}/\rho_{p}}$$(1b)−ln⁡1−Φr−Φr−ΧrΦr2=nV0(Φr/31−12Φr)(1c)$$M_{c}=\frac{\rho }{n}$$

The formula includes the swelling ratio (Φ_r_), density of natural rubber (*ρ*_p_) (0.930 g/cm^3^), density of toluene (*ρ*_s_) (0.886 g/cm^3^), molar volume of toluene (V_0_ = 106.2 mL/mol), interaction parameter between natural rubber and toluene (Χ_r_ = 0.393), and average molecular mass between cross-linking points (M_c_). The molar volume of toluene and the interaction parameters between natural rubber and toluene are considered constants, while the remaining values are experimentally determined.

## 3. Results and Discussion

### 3.1. Colloid Properties of PNRL and CPNRL

The initial solid content of pre-vulcanized natural rubber latex (PNRL) before concentration is 58%, which is insufficient for direct application in DIW 3D printing due to its high fluidity and inadequate support. This study employed the absorption concentration method based on SAP beads to increase the solid content of PNRL and regulate the rheological properties of CPNRL. CPNRL with various solid contents of 62%, 65%, 69%, 71%, and 73% can be prepared by precisely controlling the adsorption concentration time. As shown in Appendix A, CPNRL-62, CPNRL-65, and CPNRL-69 all exhibit excellent mobility. However, the mobility of CPNRL starts to reduce when the solid content reaches 71% and continues to reduce as the solid content reaches 73%, indicating that CPNRL-73 should be suitable for DIW 3D printing. The stability of PNRL, as assessed by the Zeta potential, was examined both before and after water removal using SAP (Figure 2a). The Zeta potential of PNRL was approximately −37.8 mV, while it increased to around −38.3 mV for CPNRL-73 after concentration by removing the water present in PNRL using SAP beads. The Zeta potential value serves as an indicator of the stability of PNRL [34,35]. The similar potential value of PNRL and CPNRL-73 demonstrates that concentration does not reduce the stability of CPNRL-73.

The particle size of CPNRL-73 exhibited a significant increase (Figure 2b), with the average diameter expanding from 354.6 nm to 505.3 nm. The figure illustrates that both smaller and larger rubber particles underwent an increase in size, likely attributed to the fusion and cross-linking of these particles as a result of water reduction during the concentration process. Furthermore, the SEM images revealed that the rubber particles were predominantly spherical, with smaller particles being more abundant prior to concentration, interspersed with some larger particles (Figure 2c). However, following the concentration treatment, the number of smaller particles decreased, while pear-shaped and larger particles became more prevalent (Figure 2d).

### 3.2. Rheological Properties of CPNRL

Three-dimensional printed inks, which are typically utilized in DIW printing, should demonstrate suitable rheological properties, including apparent viscosity, yield stress under both shear and compressive forces, and viscoelastic characteristics (i.e., loss and elastic moduli) [36,37,38]. The results of the rheological test of CPNRL are presented in Figure 3a,b. As depicted in the figure, the viscosity of CPNRL is influenced by shear strain, indicating the significance of enhancing the rheological properties of printing ink for improved quality. High viscosity hinders the proper extrusion and shape retention of printed samples after extrusion from the nozzle under external stress. This phenomenon demonstrates the shear thinning behavior characteristic of non-Newtonian fluids, where fluid viscosity decreases with increasing shear stress [39]. Additionally, it is evident that when the solid content reaches 71%, the CPNRL ink exhibits excellent shear thinning ability, initially increasing and then declining. In the shear modulus diagram (Figure 3b), it can also be observed that G′ > G″ for CPNRL with solid contents of 71% and 73% under low shear stress conditions. However, under higher shear forces, G′ < G″ demonstrates remarkable performance in terms of shear thinning behavior. Consequently, we conclude that CPNRL ink with a solid content of 73% or higher possesses favorable characteristics for 3D printing applications, aligning well with the rheological properties of inks required by DIW 3D printing.

### 3.3. The Stability of Line Printing, the Stacking of CPNRL Ink, and the Analysis of Scale Accuracy

The fundamental principle of DIW 3D printing to create a self-supporting extrusion layer is directly dependent on the printability and shape fidelity of the ink [38,40]. Machine parameters, such as nozzle diameter and printing velocity, significantly influence the accuracy and resolution of the printed output [41]. Typically, a nozzle with a smaller diameter can enhance printing resolution. However, this necessitates higher extrusion pressure and extended printing time to prevent nozzle clogging. Likewise, reduced print speeds generally lead to improved shape tolerances and fidelity, although this also prolongs the printing process [38].

In order to determine optimal printing parameters, the CPNRL-73 ink was subjected to testing under different printing pressures and speeds. The printing results under various combinations of pressure and speed are shown in Figure 4a,b. To ascertain the optimal printing parameters, the CPNRL-73 ink was evaluated across a range of extrusion pressures and printing velocities. Figure 4a illustrates the line width at varying speeds from 100 to 2000 mm·min^−1^, with a constant extrusion pressure of 45 kPa. The average line width of the CPNRL-73 ink is approximately double the nozzle diameter at a speed of 100 mm·min^−1^. The line width aligns closely with the nozzle diameter within the range of 500 to 1000 mm·min^−1^. Figure 4b illustrates the line width of the ink within the pressure range of 35 to 55 kPa, with increments of 5 kPa and a constant speed of 1000 mm·min^−1^. The CPNRL-73 ink can be smoothly extruded from the nozzle under a pressure of 35 kPa. In contrast, when the pressure reaches 50 kPa, the average printing line width exceeds twice the nozzle diameter. A slow line speed or excessive pressure results in excessive ink extrusion, which hinders the formation of a stable support structure, making it challenging to meet the requirements for multi-layer printing. At a printing speed of 1500 mm·min^−1^, the lines exhibit excellent form, with their width being precisely three-quarters (75%) of the nozzle diameter. Consequently, after thorough consideration, we determined that a pressure range of 45 kPa, in conjunction with a constant linear velocity of 1500 mm·min^−1^, constituted the optimal printing conditions.

The shrinkage of a specimen is inevitable when using hydrogel or rubber latex ink for 3D printing [28]. The shrinkage phenomenon of 3D-printed specimens based on the CPNRL-73 ink is also observed in this study. To evaluate the accuracy of the 3D-printed samples, we selected a solid rubber cube as a model. The inserted image in Figure 4c illustrates the XYZ orientation of the solid rubber cube after vulcanization. As depicted in Figure 4c, the dimensional fidelity of the vulcanized samples was 95.3%, 91.6%, and 96.6% in the XYZ direction, respectively. There was a decrease in volume for the printed rubber cube after vulcanization. Prior to vulcanization, the CPNRL-73 ink still contains some water, which evaporates during the vulcanization process due to heating. This results in the overall shrinkage of the structure and subsequently reduces the volume of the 3D-printed specimens.

### 3.4. Mechanical Properties

To compare the mechanical properties of the 3D-printed rubber specimens with different orientations and those prepared using the traditional casting method, a standardized dog bone tensile model was created through CAD modeling and printed by using CPNRL-73. The results indicate that different printing orientations have certain effects on the mechanical properties of the 3D-printed rubber specimens. For instance, when printed at 0°, only a strain of 720% and a stress of up to 22 MPa can be achieved. However, as the printing angle changes, there is a significant increase in stress for the samples printed at 45° and 90°. Simultaneously, the tensile strength at 90° and the elongation at break can achieve 26 MPa and approximately 800%, respectively. This phenomenon may be attributed to the anisotropic nature of the material. An increase in the angle relative to the direction of the applied force results in a reduction in the sample’s strength, which is particularly evident in polymeric materials [29]. When printed at a 90° raster angle, the contact area between the lines is maximized, and the lines align with the direction of the applied force during stretching. Conversely, the mechanical properties at 0° and 45° grating angles tend to diminish due to the misalignment between the angle and the direction of the applied force [29]. Additionally, compared to traditional casting films, the 3D-printed models exhibit higher cross-linking density across different printing angles. This discrepancy may be attributed to the lower vulcanization temperature used in traditional casting methods, leading to a reduced cross-linking density post-vulcanization compared to 3D-printed models.

Owing to the excellent interfacial compatibility between the printing layers (as shown in Figure 5), the mechanical properties of the 3D-printed rubber specimens are markedly superior to those reported in other studies (Figure 6f) [1,24,25,26,27,28,42]. The surface and cross-sectional morphology of the 3D-printed rubber specimen were examined using SEM. Distinct linear structures are evident on the surface of the rubber specimen, suggesting it was printed and fabricated layer by layer. However, no distinct interlayer interface was observed in the cross-section of the rubber specimen, indicating that excellent cross-linking between layers was formed during the vulcanization process and demonstrating a high degree of uniformity and consistency. Furthermore, the mechanical properties of the 3D-printed rubber specimen at various angles were evaluated. The device printed at 90° exhibited the highest tensile strength at 26 MPa, whereas the device printed at 0° demonstrated a strength of 22 MPa, indicating only a slight reduction. This suggests that the cross-linking between layers remained high during the vulcanization process, preventing significant delamination and preserving the superior mechanical properties of the 3D-printed rubber specimen.

### 3.5. The 3D Structures Fabricated Using CPNRL-73 Ink

Four distinct structural models were designed using CAD software and printed with the CPNRL-73 ink: a letter model, a tire tread model, rotary-shaped components, and an air rubber bag. In Figure 7, the computer-generated model, the un-vulcanized rubber specimens, and the vulcanized rubber specimens are displayed from left to right. The un-vulcanized printed specimens appear white in color because of the presence of water in the CPNRL-73 ink, and they become light yellow after vulcanization and drying by heating. The surface of the 3D-printed tire tread model was printed according to a 100% infill density, indicating that the ink was smoothly squeezed out during accumulation without any residual ink around the nozzle. The remaining 3D-printed models were stacked by rubber lines, and no collapse or inclination was observed, proving that the CPNRL-73 ink has good support and stability. Following vulcanization, the product exhibits enhanced precision with a reduced rate of shrinkage and deformation. The printed rubber product retains a shape nearly identical to the original design model.

## 4. Conclusions

In this study, we developed a new type of PNRL ink using the absorption concentration method based on SPA beads for DIW 3D printing without using additives. The total solid content of the CPNRL ink is up to 73%, and the resulting ink demonstrates good colloidal stability and 3D printability, proving that the absorption concentration method is an effective approach to fabricating rubber latex ink for DIW 3D printing. The optimal extrusion pressure was found to be approximately 45 kPa, while the optimal extrusion speed was determined to be 1500 mm/min. Furthermore, the tensile strength of the printed specimen reached an impressive 26.4 MPa when printed in the 90°direction, accompanied by a strain exceeding approximately 800%, which surpasses the majority of rubber latex specimens produced by 3D printing. The superior mechanical properties may be derived from excellent interface compatibility between the printing layers. Additionally, a wide variety of complex elastic structures with high precision can be 3D-printed by using CPNRL ink, demonstrating excellent printing adaptability and support. This study not only develops a scalable method to fabricate rubber latex ink for DIW 3D printing but also offers fresh insights into processing and structural design possibilities for complex and multifunctional rubber latex products.

## Figures and Tables

**Figure 1 polymers-17-00351-f001:**
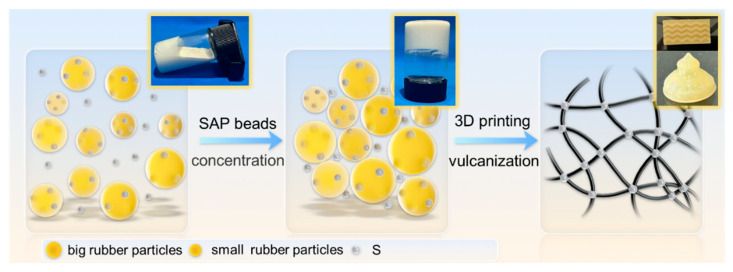
Schematic illustration of the preparation of CPNRL ink using SAP beads and a vulcanization diagram.

**Figure 2 polymers-17-00351-f002:**
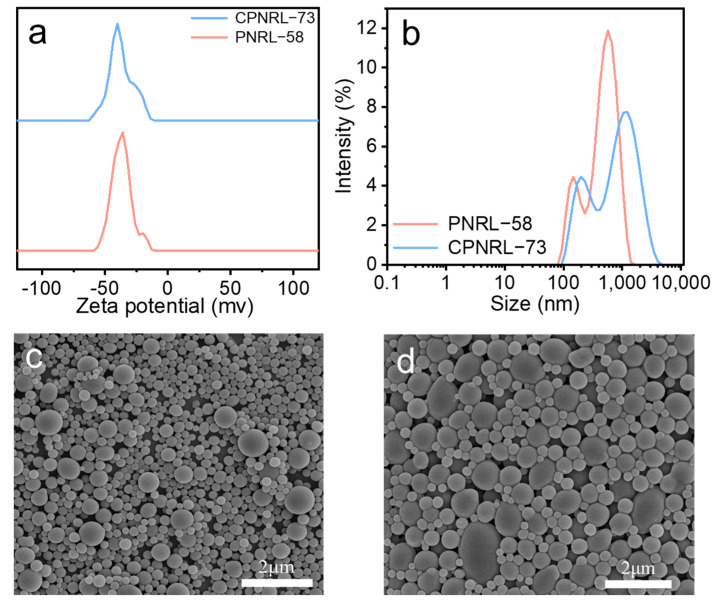
Characterization of the colloidal properties and morphology of PNRL-58% and CPNRL-73%: (**a**) Zeta potential, (**b**) particle size, (**c**,**d**) SEM images.

**Figure 3 polymers-17-00351-f003:**
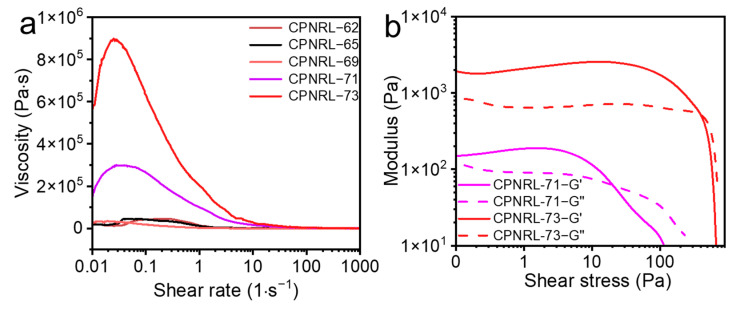
(**a**) Viscosity as a function of the shear rate of different CPNRL samples. (**b**) The correlation between modulus and shear stress of CPNRL suitable for 3D printing.

**Figure 4 polymers-17-00351-f004:**
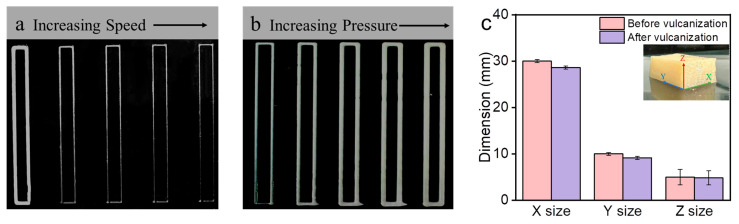
Characterization of the printing line drawings and accuracy of the CPNRL-73 ink. (**a**) At a constant pressure of 45 kPa across various speeds. (**b**) At a constant speed of 1000 mm/s under different pressures. (**c**) The dimension of the accuracy of the cubic sample in the XYZ direction after vulcanization.

**Figure 5 polymers-17-00351-f005:**
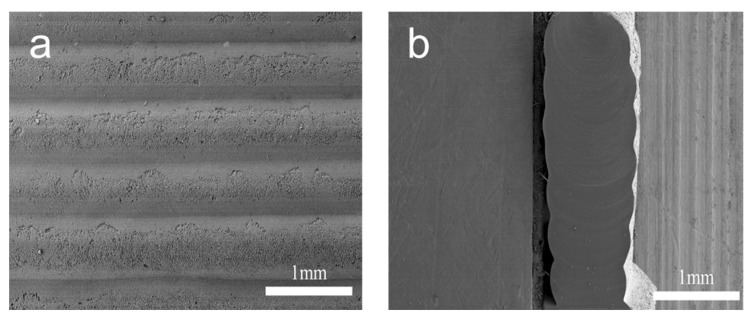
SEM images of the cross-section and surface of the 3D-printed rubber specimen. (**a**) Surface topography. (**b**) Cross-section morphology.

**Figure 6 polymers-17-00351-f006:**
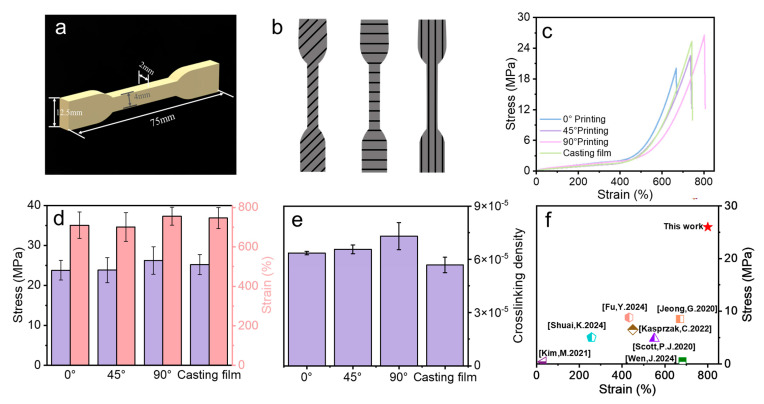
Mechanical testing and characterization. (**a**) Drawing and dimensions of CAD-generated dog bone. (**b**) Diagrams of the models for the tensile test prepared in various printing orientations. (**c**) Stress–strain curves for specimens printed in different directions. (**d**) Comparison diagram of stress–strain. (**e**) Cross-linking density of specimens printed in different directions and prepared using the casting method. (**f**) A comparative analysis of the mechanical properties of various 3D-printed elastomers [1,24,25,26,27,28,42].

**Figure 7 polymers-17-00351-f007:**
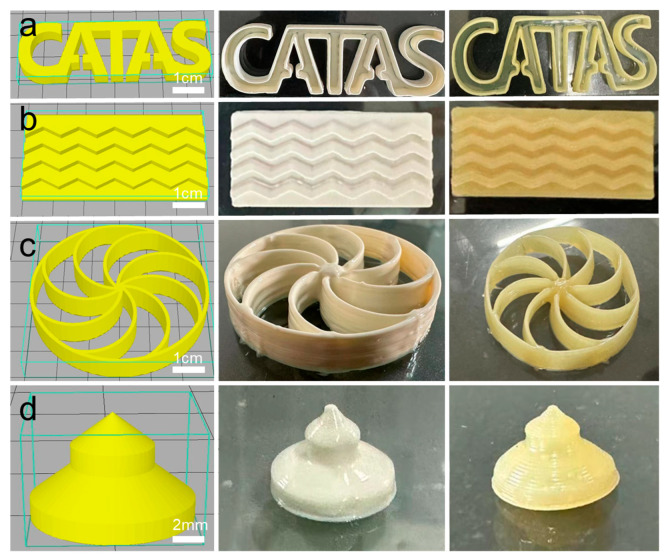
Demonstration of different 3D-printed shapes before and after vulcanization. (**a**) Alphabetic model. (**b**) Tread model. (**c**) Special-shaped part model. (**d**) Air rubber bag.

## Data Availability

The original contributions presented in this study are included in the article/Appendix A. Further inquiries can be directed to the corresponding author.

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
