# Peer review of "Concentrated Pre-Vulcanized Natural Rubber Latex Without Additives for Fabricating High Mechanical Performance Rubber Specimens via Direct Ink Write 3D Printing"

_polymers, 2025, doi:10.3390/polym17030351_

Round 1
Reviewer 1 Report
Comments and Suggestions for Authors
It is an impressive piece of work. However, some details are missing and should be clarified:
- Lines 57-58: I initially thought SAP was an additive acting as a thickening agent by increasing the viscosity of the rubber latex. Therefore, the statement and the title referring to "without additives" in this work may not be appropriate. Please reconsider this description.
- Chemical Names: Several chemical names are not disclosed in this manuscript (e.g., WLS, substance of dispersion, PX). Providing more detailed information about these chemicals, such as their types or roles, would help convince readers that the higher-performance rubber specimens can be reproduced.
- Figure 1 and SAP Beads: What does the symbol for SAP beads represent in Figure 1? Does "S" refer to sulfur linkages? This should be clarified. Additionally, where is SAP located within the rubber matrix, and what specific role does it play?
- Line 94 - Solid Content of CPNRL: The phrase "changing a period of absorption concentration" is unclear. Does this refer to the duration of an activity or the concentration of a substance? Please rephrase for clarity. Furthermore, check the grammar in this sentence to improve readability.
The paper is well written in overall, but there are still some sentences unclear.
Author Response
Comments 1: Lines 57-58: I initially thought SAP was an additive acting as a thickening agent by increasing the viscosity of the rubber latex. Therefore, the statement and the title referring to "without additives" in this work may not be appropriate. Please reconsider this description.
Response 1: Thank you for your suggestion. It is important to clarify that the viscosity of rubber latex gradually increases due to reduction of water, which is absorbed by the SAP beads during concentration process. However, the SAP beads are separated from rubber latex after concentration, which will not continue to exist in rubber latex. Therefore, we think the title "without additives" is appropriate.
Comments 2: Several chemical names are not disclosed in this manuscript (e.g., WLS, substance of dispersion, PX). Providing more detailed information about these chemicals, such as their types or roles, would help convince readers that the higher-performance rubber specimens can be reproduced.
Response 2: We appreciate your suggestions and have incorporated the specific role and classification of each substance in the supplementary materials of the article to facilitate the replication or reference by future readers in supporting information. (Page 2)
Comments 3: Figure 1 and SAP Beads: What does the symbol for SAP beads represent in Figure 1? Does "S" refer to sulfur linkages? This should be clarified. Additionally, where is SAP located within the rubber matrix, and what specific role does it play?
Response 3: Thank you for highlighting the issue. We would like to provide a more detailed explanation of Figure 1. In this figure, SAP beads serves as an adsorbent for absorbing water and are subsequently removed through filtration, which will not continue to remain in the rubber latex. The "S" you referred to is the cross-linking agent introduced during the preparation of pre-vulcanized natural rubber latex. It is important to emphasize that we use pre-vulcanized natural rubber latex rather than regular natural rubber latex. Only by utilizing pre-vulcanized natural rubber latex can rubber particles be chemically cross-linked to form a three-dimensional network, resulting in 3D-printed rubber specimens with superior mechanical properties.
Comments 4: Line 94 - Solid Content of CPNRL: The phrase "changing a period of absorption concentration" is unclear. Does this refer to the duration of an activity or the concentration of a substance? Please rephrase for clarity. Furthermore, check the grammar in this sentence to improve readability.
Response 4: Thanks for the suggestions. We have changed it to the following content “By regulating the duration of the concentration process, CPNRL with a solid content ranging from 62% to 73% can be achieved. (Page 3, Line 97)
Comments on the Quality of English Language
Point 1: The paper is well written in overall, but there are still some sentences unclear.
Response 1: We appreciate your valuable feedback. To enhance the quality of the article, we invited three renowned experts in the field of rubber to review and refine the content. The revisions have been highlighted in red for the convenience of subsequent readers.
Again, thank you very much for the comments and suggestions. If there are any problems or questions about our manuscript, please feel free to contact us.
Yours sincerely,
Jinlong Tao
On behalf of all the co-authors
Reviewer 2 Report
Comments and Suggestions for Authors
This study focuses on the application of concentrated pre-vulcanized natural rubber latex (CPNRL) as an ink (filament) for direct ink writing (DIW) in 3D printing. The research addresses challenges associated with the high fluidity and insufficient support of traditional rubber latex inks by optimizing their rheological properties. Using SAP beads for concentration, the resulting ink achieves a high solid content of 73% without the need for additives. The study highlights the mechanical performance of 3D-printed rubber specimens, with tensile strength reaching 26 MPa and strain exceeding 800% at a 90° printing angle.
Although the paper claims to have developed a new type of ink (it would be more accurate to say filament), it is essentially a modification of an existing material with adjusted properties. The experimental design does not fully support the hypothesis, as the results are not compared with reference sample(s). The reference sample should be based on previously (or commonly) used materials, or the comparison between PNRL and CPNRL should be pointed out, with the variation of parameters relevant for the research. A comprehensive comparison of the results could highlight the advantages and contributions of using CPNRL for DIW.
A serious limitation of the experimental part is the lack of reference sample in the specified context, which is an essential component of scientific soundness and needs to be corrected.
Additionally, it is not entirely clear on what basis the optimal parameters mentioned in the conclusion were defined:
“The total solid content of the CPNRL ink is up to 73%, and the resulting ink demonstrates good colloidal stability and 3D printability, proving that absorption concentration method is an effective approach to fabricate rubber latex ink for DIW 3D printing. The optimal extrusion pressure is found to be approximately 40-50 kPa, while the optimal extrusion speed was determined to be 1500 mm/min.” (lines 319 – 324)
Some results are unclear and difficult to understand (e.g., Figure 2, Figure 5).
The English language is appropriate an understandable, however, some expressions require clarification or using another term - for example, in lines 170–171: “Zeta potential of PNRL was approximately -37.8 mV, while it increased to around -38.3 mV for CPNRL-73 after concentration.” - The term “concentration” cannot stand alone as it is unclear what it refers to.
Line 43–44 contains an ambiguous sentence.
Tables S1 and S2 should be included in the main body of the article since they are directly referenced in the text.
Lines 262–265:
“This phenomenon may be attributed to the anisotropic nature of the material. An increase in the angle relative to the direction of the applied force results in a reduction in the sample's strength, which is particularly evident in polymeric materials [31].” – This explanation is not clearly connected to the study's results.
Overall, the experimental part of the manuscript lacks scientifically soundness and requires substantial and fundamental improvement.
Author Response
Comments 1: A serious limitation of the experimental part is the lack of reference sample in the specified context, which is an essential component of scientific soundness and needs to be corrected.
Response 1: Thank you very much for your suggestion. We have developed a kind of rubber emulsion ink with high mechanical properties through careful design and comprehensive characterization, which expanded the processing method of rubber emulsion ink based on 3D-printing, and provided a new idea for the preparation of rubber materials with complex structure and high strength.We carried out the experiment in a step-by-step manner. Each step of the experiment was carefully designed, and the experimental data were comprehensively characterized and analyzed, which effectively supported the research results of this paper. Firstly, we prepared a printable rubber latex ink using SAP beads, and analyzed the colloidal properties of ink in detail; after that, we studied the rheological properties of the ink, and found that CPNRL-73 is most suitable for DIW 3D printing; next, we studied the influence of extrusion pressure and linear velocity on Printing, and found the best extrusion pressure and linear speed; finally, we studied the effect of different printing angles on the mechanical properties of the printed specimens, and proved that the 3D-printed product had the best mechanical properties when printing at 90 degrees. In addition, we also compared the mechanical properties of 3D-printed specimens and the casting film, indicating DIW 3D printing retained good mechanical properties.
Comments 2: Additionally, it is not entirely clear on what basis the optimal parameters mentioned in the conclusion were defined: “The total solid content of the CPNRL ink is up to 73%, and the resulting ink demonstrates good colloidal stability and 3D printability, proving that absorption concentration method is an effective approach to fabricate rubber latex ink for DIW 3D printing. The optimal extrusion pressure is found to be approximately 40-50 kPa, while the optimal extrusion speed was determined to be 1500 mm/min.” (Lines 319 – 324)
Response 2: Thanks for your question. The optimal parameters were determined through the design of a rectangular model, systematically varying extrusion pressures and printing speeds. Figures 4a and 4b in the paper illustrate the line width of the ink extrusion under different pressures and speeds. Based on the actual printing requirements, the optimal printing speed and pressure were determined to be 1500 mm/min and 45 kPa, respectively. Upon review, it was noted that the statement regarding 40-50 kPa was incorrect, and this has been corrected.
Comments 3:The English language is appropriate an understandable, however, some expressions require clarification or using another term - for example, in lines 170–171: “Zeta potential of PNRL was approximately -37.8 mV, while it increased to around -38.3 mV for CPNRL-73 after concentration.” - The term “concentration” cannot stand alone as it is unclear what it refers to.
Response 3: Thank you very much for your advice. We have modified it as follows: “The stability of PNRL, as assessed by Zeta potential, was examined both before and after water removal using SAP. Zeta potential of PNRL was approximately -37.8 mV, while it increased to around -38.3 mV for CPNRL-73 after concentration by removing water present in PNRL using SAP beads.” ( Page 4, Lines 177-181)
Comments 4: Line 43–44 contains an ambiguous sentence.
Response 4: Thank you very much for your suggestion. We apologize for our negligence, and we have amended it “However, they also demonstrates some limitations.”
Comments 5: Tables S1 and S2 should be included in the main body of the article since they are directly referenced in the text.
Response 5: Thank you for your proposal. In this section, we only aim to present the preparation process of pre-vulcanized natural rubber latex. The table referred to is intended to enable readers to replicate our pre-vulcanized rubber latex preparation. Considering that the preparation of pre-vulcanized rubber latex is not the main research content of this paper, we still recommend to put it in the supporting information.
Comments 6: “This phenomenon may be attributed to the anisotropic nature of the material. An increase in the angle relative to the direction of the applied force results in a reduction in the sample's strength, which is particularly evident in polymeric materials [31].” – This explanation is not clearly connected to the study's results.
Response 6: Thanks for your question. We make a clarification: the anisotropy of materials in 3D printing refers to that the orientation and distribution of polymer materials are affected by the fluidity, viscosity and particle size of materials during the printing process. When subjected to shear force, polymer materials may be oriented along the extrusion direction, resulting in higher strength in the printing direction [3]. The anisotropy mentioned in line 262 means that samples with the same printing direction and force direction (90 ° grating angle) have higher mechanical properties during stretching.
Comments 7: Overall, the experimental part of the manuscript lacks scientifically soundness and requires substantial and fundamental improvement.
Response 7: Thank you for your questions. We carried out the experiment in a step-by-step manner. Each step of the experiment was carefully designed, and the experimental data were comprehensively characterized and analyzed, which effectively supported the research results of this paper.The experimental part mainly involves the following contents: First, pre-vulcanized natural rubber latex (solid content 58%) was prepared. Secondly, we concentrated the pre-vulcanized natural rubber latex by super absorbent beads to obtain concentrated pre-vulcanized natural rubber latex with different solid content (solid content from 62% to 73%). Then, determined which solid content of concentrated pre-vulcanized latex is suitable for 3D printing, and the results show that CPNRL-73 is the most suitable for DIW 3D printing. Next, for comparison, the mechanical properties of the rubber parts printed by CPNRL-73 and its casting film were compared, the 3D-printed rubber parts showed slightly higher mechanical properties. Finally, CPNRL-73 ink can be used to print different complex structures, indicating that it has good printing performance, which provides a new idea for building high-strength and complex rubber products.
Again, thank you very much for the comments and suggestions. If there are any problems or questions about our manuscript, please feel free to contact us.
Yours sincerely,
Jinlong Tao
On behalf of all the co-authors
Reviewer 3 Report
Comments and Suggestions for Authors
The paper presents an innovative rubber ink intended for direct ink writing. The ink is characterized to show its suitability for printing and the final parts display remarkable mechanical properties compared to other 3D-printed rubbers. Unfortunately, I have serious issues with the citations used at the end of the introduction and some parts of the text are inconsistent with the figures. Please see below my recommendations:
Line 38: Reference 13 is not about natural rubber.
Line 41: References 22-23 do not use natural rubber, reference 24 is a blend using natural rubber and not pure natural rubber, and reference 25 is not FDM. Please change the text and references accordingly.
Line 42: References 1, 26-27 and 29 do not use natural rubber.
Line 43: Reference 32 does not use natural rubber
Line 51: The conclusion of the added value of DIW does not appear natural to me given the previous sentences.
Line 63: What do you mean by "alerting printing line angle"?
Line 85: What are the two (group of) components used to compute the mass fraction?
Line 86: Does the filtering preserve the composition of the dispersion?
Line 98: If possible, please add an illustration to help visualize the printed geometries.
Line 100: What is the width of the rectangular model?
Lines 102-104: What is the difference between the linear speeds of 100-2000 mm/min and 1000 mm/min and between the pressures of 35-55 kPa and 45 kPa? Please clarify.
Line 113: Did you ensure that the filtering does not alter the solid content of the CPNRL?
Line 119: Which open-source software did you use?
Lines 145-150: Please explicit the symbols rho_p, and n.
Line 151: How are the values of the parameters determined?
Line 172: Reference 10 does not discuss the effect of zeta potential on the colloidal stability of PNRL. Please change for an appropriate reference.
Lines 176-183: The results from the Zeta sizer suggest that both populations (small and big particles) grow in size during the concentration. However, the text does not reflect this result. I suggest adapting the text accordingly.
Line 184: Does the filling of the gaps exist in the actual samples or is it an artefact of SEM?
Lines 201-203: In the text, you state that G'<G" at low shear stress and G'>G" at high shear stress but Figure 3b shows the opposite. Please check what part is correct and change the other accordingly.
Line 203: For CPNRL-71 at high shear stress, is the difference between G' and G" significant given the uncertainties of the measurements?
Line 206: What is the characteristic shear stress in DIW? Are the shear-thinning behaviour of the rubber coherent with this stress?
Line 223: Only the velocity range is given. What are the individual velocities?
Line 224: The pressure chosen to explore the influence of the printing speed is orders of magnitude higher than the pressures used for the rest of the experiments. Please explain the choice of pressure.
Line 230: The text states that the line width increases with an increase in pressure but Figure 4b suggests the opposite. Please clarify the findings.
Line 235: What are the arguments for choosing a range of pressure and not a precise pressure?
Line 262: Please clarify what is the "anisotropic nature of the material". The change in the mechanical properties of 3D-printed parts comes often from a weaker bond between the layers, which is different from an anisotropy at the molecular level.
Line 270: By which phenomenon does the angle influence the crosslinking density? It is not clear to me how a macroscopic parameter such as the contact length between the lines can influence crosslinking at the molecular scale.
Line 273: I assume that the last sentence applies to the 0° angle. Please make clear in the text what angle is used to draw this last conclusion.
Line 293: In Figure 6f, the points for references 1 and 30 seem different from the values reported in the paper. Also, the stress for reference 27 seems negative. Please check carefully these three points.
Line 23: Either the strain exceeds 800% or it is approximately 800%, please choose one of the two formulations.
Line 44: The sentences "Although they..." and "However, they ..." should be combined into one to improve the readability.
Line 65: The syntax of "accompanied by a strain reached approximately 800%" seems problematic.
Line 99: Missing hyphen between 3D and printed.
Line 128: The "respectively" seems odd. Please check the sentence.
Line 144: Remove "Formula 1" from the text.
Author Response
Comments 1: Line 38: Reference 13 is not about natural rubber.
Response 1: We appreciate your valuable feedback, and deeply sorry for our negligence. After verification, we confirmed that reference 13 is not pertinent to the topic of natural rubber, and therefore it has been removed. (Page 1, Line 38)
Comments 2: Line 41: References 22-23 do not use natural rubber, reference 24 is a blend using natural rubber and not pure natural rubber, and reference 25 is not FDM. Please change the text and references accordingly.
Response 2: Thank you for your meticulous review. The passage from Line 41 was intended to describe the current predominant method of 3D printing rubber elastomers. Due to an oversight on my part, there was an inaccuracy that led to a significant error. We have revised "natural rubber" to "rubber elastomer" on Lines 41-60 to ensure greater precision. Additionally, reference 25 does not belong to FDM 3D printing, and it has been deleted. (Page 1, Lines 41-60)
Comments 3: Line 42: References 1, 26-27 and 29 do not use natural rubber.
Response 3: Thanks for your careful review. We have adjusted “natural rubber” to “rubber”, simultaneously adjusted the references.
Comments 4: Line 43: Reference 32 does not use natural rubber
Response 4: Thank you for your meticulous review. We have adjusted “natural rubber” to “rubber elastomer”, simultaneously adjusted the references.
Comments 5: Line 51: The conclusion of the added value of DIW does not appear natural to me given the previous sentences.
Response 5: Thank you for pointing out the problem, which is very helpful to us. We have modified Lines 48-56 as follows: Regarding the DIW technique, it is necessary to regulate the rheological properties of the ink, for instance, by introducing rheological modifiers to satisfy specific requirements. In comparison with the aforementioned two methods, DIW 3D printing exhibits lower costs and simpler operational procedures. In summary, compared to FDM and VPP, DIW 3D printing of rubber elastomers is regarded as one of the most promising technologies. It eliminates the need for high temperatures and photosensitizers while enabling the incorporation of a wider range of additives in 3D printing inks. However, the addition of substances such as rheological regulators may result in increased costs and volume shrinkage of printed samples. Moreover, these additives may compromise the mechanical properties of the printed objects. Consequently, there is an urgent need for an additive-free and scalable method to manufacture rubber samples with superior mechanical properties through DIW 3D printing.”(Page 2, Line 47-59)
Comments 6: Line 63: What do you mean by "alerting printing line angle"?
Response 6: Thank you for your valuable suggestion. We have replaced the term "alerting printing line angle," with "raster orientation" to enhance the professionalism and technical accuracy of the article. (Page 2, Line 66)
Comments 7: Line 85: What are the two (group of) components used to compute the mass fraction?
Response 7: Thank you for your inquiry. We employ the gravimetric method to determine the solid content, which involves comparing the mass ratios of the dispersion before and after water evaporation.
Comments 8: Line 86: Does the filtering preserve the composition of the dispersion?
Response 8: Thank you for your question. The filtration of dispersion is a common treatment method used in the preparation of pre-vulcanized natural rubber latex, mainly to effectively remove bubbles generated during the stirring process, without affecting the composition of the dispersion.
Comments 9: Line 98: If possible, please add an illustration to help visualize the printed geometries.
Response 9: Thank you for your suggestion. The model of printed geometries have been added to the supporting information. (Page 4)
Comments 10: Line 100: What is the width of the rectangular model?
Response 10: Thank you for your valuable question. The rectangular model with a length of 20 cm and a width of 1cm has been added to the supporting information. (Page 4)
Comments 11: Lines 102-104: What is the difference between the linear speeds of 100-2000 mm/min and 1000 mm/min and between the pressures of 35-55 kPa and 45 kPa? Please clarify.
Response 11: Thank you for your question. The variation of printing speed and pressure vary affects the extrusion line width. To determine the optimal parameters, a speed range of 100-2000 mm/min and an extrusion pressure range of 35-55 kPa are explored to identify the most suitable settings within these intervals.
Comments 12: Line 113: Did you ensure that the filtering does not alter the solid content of the CPNRL?
Response 12: Thank you for your valuable feedback. We apologize for any confusion caused by our oversight. The filtration is employed to eliminate bubbles, and the solid content of CPNRL remains unchanged before and after the filtering. Additionally, we added a description of the preparation of casting rubber film in the article. (Page 3, Lines 115-118)
Comments 13: Line 119: Which open-source software did you use?
Response 13: Thank you for your proposal. We used Slic3r software and also modified it. (Page 3, Line 124)
Comments 14: Lines 145-150: Please explicit the symbols rho p, and n.
Response 14: We appreciate your advice. The symbols in lines 149-150 have been revised.
Comments 15: Line 151: How are the values of the parameters determined?
Response 15: Thank you for your valuable suggestions. We have explained the source of the value. The molar volume of toluene and the interaction parameters between natural rubber and toluene are considered constants, while the remaining values are experimentally determined. (Page 4, Lines 155-160)
Comments 16: Line 172: Reference 10 does not discuss the effect of zeta potential on the colloidal stability of PNRL. Please change for an appropriate reference. (Page 4, Line 181)
Response 16: Thank you for your valuable comment. We have deleted reference 10 and add reference [31-32] which discussed the effect of zeta potential on the colloidal stability of rubber latex.
Comments 17: Lines 176-183: The results from the Zeta sizer suggest that both populations (small and big particles) grow in size during the concentration. However, the text does not reflect this result. I suggest adapting the text accordingly.
Response 17: Thank you for your valuable suggestions. We have adapted the text. (Page 5, Line 183-191)
Comments 18: Line 184: Does the filling of the gaps exist in the actual samples or is it an artefact of SEM?
Response 18: We appreciate your valuable feedback. After thorough discussion, we acknowledge that this phenomenon is indeed challenging to differentiate rigorously. Therefore, we have decided to remove the sentence at Line 184.
Comments 19: Lines 201-203: In the text, you state that G'<G" at low shear stress and G'>G" at high shear stress but Figure 3b shows the opposite. Please check what part is correct and change the other accordingly.
Response 19: Thank you for pointing out the problem. Due to our negligence, there are serious errors in the description in the article. We are very sorry for this, and have modified it as follows: it can also be observed that G'>G'' for CPNRL with solid contents of 71% and 73% under low shear stress conditions. However, under higher shear forces, G'<G'' demonstrates remarkable performance in terms of shear thinning behavior. (Page 5, Line 207 and 209)
Comments 20: Line 203: For CPNRL-71 at high shear stress, is the difference between G' and G" significant given the uncertainties of the measurements?
Response 20: Thank you for your valuable suggestions. The difference is obvious between G' and G" for CPNRL-71 at high shear stress. Although the viscosity of CPNRL-71 also meets the requirements of ink for DIW 3D printing, because the formability is poor after 3D printing, and the subsequent vulcanization process result in high shrinkage of 3D-printed specimens, we select CPNRL-73 as the optimal ink.
Comments 21: Line 206: What is the characteristic shear stress in DIW? Are the shear-thinning behaviour of the rubber coherent with this stress?
Response 21: Thank you for your inquiry. After reviewing the relevant information, we would like to provide an explanation regarding the DIW printing process. During this process, the ink experiences varying levels of shear stress as it is extruded through the nozzle. Inside the nozzle, the shear stress exceeds the yield stress, causing the material to yield and flow. Once outside the nozzle, the removal of stress allows the material to transform into a viscoelastic solid. Following the deposition process, the printed structure must maintain its shape until full solidification occurs. Post-printing, the ink's rheological properties should be optimized to prevent sagging of stacked filaments due to their own weight and to minimize bending in span parts. To mitigate these issues, the storage modulus (G') of the ink should exceed the loss modulus (G''). Additionally, for DIW inks, the G' value should remain nearly constant under low shear stress conditions, known as the linear viscoelastic region (LVR). Two yield stress parameters (σ1 and σ2) can be optimized to better understand the ink's behavior during printing: 1) σ1, which marks the onset of a decline in the G' value (deviation from LVR), and 2) σ2, which signifies the transition from solid-like to liquid-like behavior (cross-over point of G' and G'') [2]. As evidenced by the rheological diagram, the behavior of our CPNRL-73 ink aligns with the aforementioned conditions and exhibits shear stress characteristics consistent with those observed in DIW.
Comments 22: Line 223: Only the velocity range is given. What are the individual velocities?
Response 22: Thank you for your question. The specified speed and pressure ranges are designed to facilitate the identification of the optimal printing parameters. The individual velocities have not been specified.
Comments 23: Line 224: The pressure chosen to explore the influence of the printing speed is orders of magnitude higher than the pressures used for the rest of the experiments. Please explain the choice of pressure.
Response 23: Thank you for bringing this issue to our attention. We sincerely apologize for the error in parameter writing that occurred due to my oversight. The incorrect value has been corrected from 1000 kPa to 45 kPa on line 229. (Page 6, Line 230)
Comments 24: Line 230: The text states that the line width increases with an increase in pressure but Figure 4b suggests the opposite. Please clarify the findings.
Response 24: Thank you for pointing out the problem. After investigation, it is because the pressure increases from right to left when printing, but it is not processed when making pictures, which leads to deviation in consistency. We have made a profound reflection on this, and modified it. (Page 7, Line 254)
Comments 25: Line 235: What are the arguments for choosing a range of pressure and not a precise pressure?
Response 25: Thanks for your question. Owing to an inadvertent omission in the documentation, the specified pressure was not detailed in this section. Additionally, please be informed that the pressure for Line 230 has been revised to 45 kPa.
Comments 26: Line 262: Please clarify what is the "anisotropic nature of the material". The change in the mechanical properties of 3D-printed parts comes often from a weaker bond between the layers, which is different from an anisotropy at the molecular level.
Response 26: Thanks for your question. We make a clarification: the anisotropy of materials in 3D printing refers to that the orientation and distribution of polymer materials are affected by the fluidity, viscosity and particle size of materials during the printing process. When subjected to shear force, polymer materials may be oriented along the extrusion direction, resulting in higher strength in the printing direction [3]. The anisotropy mentioned in the article means that samples with the same printing direction and force direction (90 ° grating angle) have higher mechanical properties during stretching.
Comments 27: Line 270: By which phenomenon does the angle influence the crosslinking density? It is not clear to me how a macroscopic parameter such as the contact length between the lines can influence crosslinking at the molecular scale.
Response 27: Thank you very much for your question. After discussion, we have reached an agreement with you that it is difficult to explain the crosslinking density using macroscopic parameter. Therefore, we have modified the statement as follows: When printed at a 90° raster angle, the contact area between the lines is maximized, and the lines align with the direction of the applied force during stretching. Conversely, the mechanical properties at 0° and 45° grating angles tend to diminish due to the misalignment between the angle and the direction of the applied force. Additionally, compared to traditional casting films, 3D-printed models exhibit higher cross-linking density across different printing angles. This discrepancy may be attributed to the lower vulcanization temperature used in traditional casting methods, leading to a reduced cross-linking density post-vulcanization compared to 3D-printed models. (Page 7, Line 271-278)
Comments 28: Line 273: I assume that the last sentence applies to the 0° angle. Please make clear in the text what angle is used to draw this last conclusion.
Response 28: Thank you for your inquiry. This conclusion is derived from a comparative analysis of dog bone models printed at various grating angles following tensile testing. The mechanical properties of the models printed at 0° grating angle are marginally lower than those printed at 90°. We have incorporated Reference 31 to substantiate the conclusions presented.
Comments 29: Line 293: In Figure 6f, the points for references 1 and 30 seem different from the values reported in the paper. Also, the stress for reference 27 seems negative. Please check carefully these three points.
Response 29: We appreciate your identification of the issue. Following a thorough review, we have reorganized the parameters presented in Figure 6f.
Comments on the Quality of English Language
Point 1: Line 23: Either the strain exceeds 800% or it is approximately 800%, please choose one of the two formulations.
Response 1: Thanks for your suggestion. We choose “it is approximately 800%”.
Point 2: Line 44: The sentences "Although they..." and "However, they ..." should be combined into one to improve the readability.
Response 2: Thanks for your suggestion. We have modified it according to your suggestion.
Point 3: Line 65: The syntax of "accompanied by a strain reached approximately 800%" seems problematic.
Response 3: Thank you for your valuable advice. We have modified it according to your suggestion.
Point 4: Line 99: Missing hyphen between 3D and printed.
Response 4: Thank you for your valuable advice. We have modified it according to your suggestion.
Point 5: Line 128: The "respectively" seems odd. Please check the sentence.
Response 5: Thank you for your valuable advice. We have deleted it according to your suggestion.
Point 6: Line 144: Remove "Formula 1" from the text.
Response 6: Thank you for your valuable advice. We have deleted it according to your suggestion.
[1] N. H. Yusof, M. Singh, F. R. Mohd Rasdi, K. S. Tan, Journal of Rubber Research 2023, 26, 169.
[2] M. A. S. R. Saadi, A. Maguire, N. T. Pottackal, M. S. H. Thakur, M. M. Ikram, A. J. Hart, P. M. Ajayan, M. M. Rahman, Advanced Materials 2022, 34, 2108855.
[3] J. Chen, X. Liu, Y. Tian, W. Zhu, C. Yan, Y. Shi, L. B. Kong, H. J. Qi, K. Zhou, Advanced Materials 2022, 34, 2102877.
Again, thank you very much for the comments and suggestions. If there are any problems or questions about our manuscript, please feel free to contact us.
Yours sincerely,
Jinlong Tao
On behalf of all the co-authors
Round 2
Reviewer 2 Report
Comments and Suggestions for Authors
I thank the authors for the changes made in the paper. I will recommend the acceptance and publication of the proposed research.
Reviewer 3 Report
Comments and Suggestions for Authors
The authors answered all my points. The only error I see is in line 111: the "i" in "casting" is duplicated.